# Review on Extraction, Modification, and Synthesis of Natural Peptides and Their Beneficial Effects on Skin

**DOI:** 10.3390/molecules28020908

**Published:** 2023-01-16

**Authors:** Jiabing An, Ivan Stève Nguepi Tsopmejio, Zi Wang, Wei Li

**Affiliations:** 1College of Chinese Medicinal Materials, Jilin Agricultural University, Changchun 130118, China; 2College of Life Sciences, Engineering Research Center of the Chinese Ministry of Education for Bioreactor and Pharmaceutical Development, Jilin Agricultural University, Changchun 130118, China

**Keywords:** peptides, skin wounds, extraction, modification, synthesis

## Abstract

Peptides, functional nutrients with a size between those of large proteins and small amino acids, are easily absorbed by the human body. Therefore, they are seeing increasing use in clinical medicine and have revealed immunomodulatory and anti-inflammatory properties which could make them effective in healing skin wounds. This review sorted and summarized the relevant literature about peptides during the past decade. Recent works on the extraction, modification and synthesis of peptides were reviewed. Importantly, the unique beneficial effects of peptides on the skin were extensively explored, providing ideas for the development and innovation of peptides and laying a knowledge foundation for the clinical application of peptides.

## 1. Introduction

With the development and renewal of science and technology, researchers eventually discovered a class of organic compounds whose molecular weight lies between proteins and amino acids. These compounds are easily absorbed, require low energy consumption to produce, and demonstrate high affinity, specificity, and low toxicity [1]. These compounds are known as peptides, and have been revealed as new components of therapeutic drugs. An increasing number of studies have proved that peptides have unique efficacy in antibacterial, anti-inflammatory, and anti-tumor aspects [2,3]. Given their attractive pharmacological and intrinsic properties, peptides are considered an excellent starting point for the design of new therapies, with good safety, tolerability, and efficacy in clinical application [4]. This provides huge advantages over traditional small molecules. In addition, peptide-based therapies typically have a lower production complexity than protein-based biopharmaceuticals [5], which significantly reduces production costs. Therefore, in this regard, peptides are optimally positioned between small molecules and biopharmaceuticals, and given their increased use, suitable methods for efficiently extracting them from natural sources have become the focus of attention [6]. However, many studies have shown that naturally occurring peptides are generally not suitable for direct clinical application because of their inherent weaknesses [7], including poor chemical and physical stability, and short circulating plasma half-life [8]. To address these issues, researchers must conduct studies to improve the application of peptides derived from modification and synthesis. 

Although it is not fatal, skin damage often increases pain and affects the self-image of the patient; regeneration and wound healing are also essential for tissue homeostasis and the survival of organisms [9]. The causes of skin wounding are diverse, and the underlying mechanisms of wound healing are equally complex, such as inflammation [10] and oxidative stress [11]. It is well known that increasing numbers of scholars are interested in the exploration of skin diseases. Peptides have revealed many biological functions, most notably as signaling/regulatory molecules in a variety of physiological processes, including anti-inflammatory, defense, immunity, and homeostasis. These have been identified as good choices for skin healing agents [12]. This review integrated recent studies, introduced the extraction, modification, and synthesis of peptides, and focused on exploring the research progress of peptides in the field of skin wounding, providing a good knowledge foundation for the future application and development of peptides. 

## 2. Extraction of Peptides

In recent years, much attention has focused on the extraction and purification of peptides. Figure 1 shows the current basic process for obtaining peptides. The development and utilization of peptides also provide new ideas for the innovation of therapeutic drugs. To increase the peptide extraction rate, enzymolysis and pretreatment are often used before extraction and separation. There are many types of proteases in nature. Proteases can be divided into three categories according to their origin: proteases of plant origin, proteases of animal origin, and proteases of microbial origin. Papain is a highly active endo-cysteine protease from papaya. It is one of the widely used proteases of plant origin. Trypsin is an important endoprotease in human and animal intestines. In the pancreas, trypsin is produced by activating trypsinogen [13]. Flavourzyme is sold as an industrial peptide enzyme preparation derived from *Aspergillus oryzae* [14]. Proteases can also be classified according to their pH value as alkaline proteases, neutral proteases, or acidic proteases. Although all three of these proteases are found in plants and animals, microbial populations are their most widespread source [15]. Researchers have generally applied five kinds of hydrolase (Flavourzyme, trypsin, acid protease, neutral protease, and alkaline protease) to extract antioxidant peptides from the mackerel (*Scomberomorus niphoniusis*) defatted visceral powder. The diphenyl bitter hydrazine radical scavenging rate, hydroxyl radical scavenging rate, and hydrolysis degree are used as indicators for the selection and optimization of hydrolytic enzymes to optimize the best hydrolysis solution [16]. This was the case with apricot kernel (*Semen Armeniacae Amarum*) hydrolysate that was obtained by hydrolysis and degreasing with the compound protease of alkaline protease and Flavourzyme [17]. Some studies have used trypsin, Flavourzyme, and neutral and alkaline protease to extract antioxidant proteins from frog breast oil (*Ranae Oviductus*) [18].

After obtaining the crude extract, chromatography is often applied to separate the desired peptides. According to the separation principle, chromatography can be divided into adsorption chromatography, ion-exchange chromatography, gel chromatography, and distribution chromatography. Experiments have demonstrated that the cation exchange column has been widely used in the separation and purification of peptides. This is the case for the separation and analysis of active antioxidant peptides from mackerel, and it was found to be the most suitable chromatographic method in [16]. Reportedly, the cation exchange column was also developed to enrich protein N-terminal peptides. Briefly, N-terminal peptides with or without n-acetylation can be separated from internal peptides by strong cation-exchange chromatography according to the charge/orientation retention type based on the peptides [19]. Surprisingly, the separation of responsible peptides from egg white hydrolysates [20] and antioxidant peptides from feather hydrolysates [21] was optimized by cation exchange chromatography and reverse-phase chromatography. Furthermore, it was reported that purified hirudin peptides were obtained from leeches (*Hirudo*) by strong base anion-exchange column chromatography and G10 gel column chromatography [22].

Gel filtration has also been the preferred option for obtaining the desired active ingredients from the crude extract. This was the case with the hydrolysates of pearl millet (*Pennisetum glaucum*) that were separated by gel filtration chromatography to obtain antioxidant peptides [23]. The active peptides of apricot kernel (*Semen Armeniacae Amarum*) hydrolysates were further isolated by gel filtration chromatography on Sephadex G-25 and G-15, and their antioxidant potential was further evaluated and proved [17]. In order to understand the taste of Philippine clams (*Ruditapes philippinarum*), 14 novel umami peptides were isolated and identified by gel chromatography, HPLC, and UPLC-ESI-QTOF-MS/MS [24]. A peptide was also found in the foot of green mussel (*Perna canaliculus),* which was purified by size-exclusion chromatography (SEC); its sequence was identified by LC-MS/MS and its anti-inflammatory effect was investigated by in vitro experiment [25].

For the separation of peptides, HPLC has gained the highest application value because of its high speed, high efficiency, and high sensitivity [26,27]. This was evident in peptides isolated from fermented milk (yogurt) [28] and rapeseed *(Brassica campestris* L.) [29] by preparative reversed-phase HPLC, then purified and analyzed by ESI-MS/MS. Notably, a novel peptide with antiplatelet activity was isolated and identified from silver carp (*Hypophthalmichthys molitrix*) skin by the same method [30], as was a novel peptide purified by solid-phase extraction and HPLC from the interleaf of banana (*Musa paradisiaca*) plants, with its primary structure determined by MS and amino acid analysis [31].

## 3. Modification of Peptides

Natural active peptides are known to play an irreplaceable role in immune regulation [31,32], immune hormones [33], enzyme inhibition [34], and antiviral properties [35,36]. Despite their potential use as therapeutic agents, there are many potential problems with natural peptides due to their low stability and proteolysis, resulting in short activity duration and low bioavailability in vivo. One way to overcome these shortcomings is to use modified peptides, known as peptides mimics [37]. For example, natural peptides found in venoms could be used directly in routine therapy, but many of these peptides might need to be truncated or stabilized to improve their therapeutic properties. Thus, a complementary strategy is the generation of peptides mimics by displaying key residues forming the pharmacophore of the peptide toxin on a non-peptide scaffold [38]. Some studies have proposed a chemical modification box for peptides, which was used for the modification of peptides’ skeleton, amino acid side chain, and higher-order structure. This method was used to overcome the main issues encountered during the transition from natural peptides to peptide therapeutic agents, therefore promoting the synthesis and development of solid-phase peptides [39]. To improve the activity and increase the function of peptides, the NMEGylation-covalent binding of oligo-*N*-methoxyethylglycine (NMEG) chains was evaluated as a novel form of peptide/protein modification, especially for the stability and solubility of C_20_ peptides [40]. In addition, a new type of peptide was designed by a modified method, which greatly broadened the application space of peptides in different fields. To form a novel peptide, a six-membered carbon ring with an amino group on the ring binds substituted amino acids to arginine-rich peptides. Further studies found the value of this peptide in the development of cell-penetrating peptides [41]. The physicochemical properties of peptides are generally regulated by introducing one or more methyl groups into peptidyl amide bonds, while the pharmacokinetic properties of peptides are endowed with unprecedented characteristics [42].

## 4. Synthesis of Peptides

The applications of different modification methods have significantly improved the inherent shortcomings of natural peptides, such as stability and cell penetration. In addition to designing new peptides by modification, it was possible to understand the synthesis of new compounds that do not exist in Nature by using different methods and means.

Previous studies provided new ideas for the development and utilization of peptides, as well as new therapeutic directions for clinical application. In one work it was reported that a peptide was synthesized based on a known chemical formula. The basic peptide components of the *Lactobacillus casei* peptidoglycan complex were used as a reference to compose this chemical formula, which has potential as an effective anti-tumor agent [43]. A new method has been developed in which lysine residues are linked to the C-terminal of the desired peptides by a standard peptide bond during synthesis. The immobilized carboxypeptidase B (CPB) is then used to remove these lysine residues after purification, thus improving the total synthesis and purification yield of the peptides [44]. Similarly, there is a method in which the heterozygous organic peptides’ macrocyclic compounds are synthesized by cyclizing ribosomal-derived peptide sequences with non-peptide organic connectors [45]. Furthermore, cyclic RGD peptides could be efficiently synthesized based on microflow triphosgene-mediated peptide chain extension and microflow photochemical macrocyclic lactamization [46]. A novel strategy was also described for the generation of bicyclic peptides containing non-peptide skeleton elements, starting from recombinant peptide precursors. These compounds were produced by a ‘one-pot and two-step’ sequence in which the peptides were macrocycled via bifunctional oxyamine/1,3-amino-thiol synthetic precursors, and then the intramolecular disulfide was formed between the synthetic precursor mercaptan and a cysteine embedded in the peptide sequence [47]. In another one-pot method, goadsporin (GS) was synthesized using recombinant enzymes in a flexible in vitro translation system (called the FIT-GS system) [48].

## 5. Beneficial Effects of Peptides on Skin

The skin is composed of epidermis, dermis and subcutaneous tissue. Understanding skin structure is fundamental for the treatment of all skin conditions. The healing of skin wounds is an important biological process which can regenerate new skin after a wound. Skin injuries can be divided into skin trauma and burns, skin disease, and skin cancer. Among them, chronic wounds caused by skin injuries and burns are the most common skin diseases due to the slow healing of hypoxia, abnormal peripheral sensory nerve function, and insufficient blood tissue supply. The most significant sign of chronic wounds is severe abnormal immune skin function [49]. The active components of peptides could serve as first-line innate immune defense against exogenous microorganisms in the skin, in addition to coordinating adaptive immune responses to perform various immunomodulatory functions. Different authors found that peptides repair skin damage through a variety of mechanisms (Figure 2) [50,51]. Many skin diseases and injuries have been reported to involve the production of ROS radicals [52], and a dramatic increase in ROS levels can cause oxidative stress. Peptides acting on the skin can have a therapeutic effect by inhibiting the production of ROS. In addition, the skin has a vast antioxidant system, including superoxide dismutase (SOD), glutathione peroxidase (GPX) and catalase (CAT) [53], and the therapeutic process of peptides on the skin involves the regulation of these factors. When skin pathology occurs, it is often regulated by the PI3K/AKT [54], MAPK/ERK [55], and TGFβ/Smad pathways [56]. Further studies have shown that peptides can regulate inflammatory factors (IL-1, IL-6, IL-8) or matrix metalloproteinases (MMP1, MMP2, MMP3) by PI3K/AKT, MAPK/ERK, and TGFβ/Smad pathways, thereby reducing the inflammatory response of the skin [57,58].

### 5.1. Skin Burns and Trauma

There are many kinds of chronic skin wound disorders, such as the commonly observed skin burns [59,60] and trauma [61]. Researchers have discovered many medicines [62] and methods [63,64] that can treat chronic wounds. According to relevant data, effective peptides in skin injury have been widely extracted and found in Nature [65]. With the innovation and development of science and technology, scientists have gradually explored the specific mechanism of those peptides for skin therapy. Traditional herbal products are often applied for skin healing, and plant therapeutic agents such as honeysuckle (*Lonicera japonica Thunb*) [66], patchouli (*Pogostemon cablin (Blanco) Benth*) [67], and aloe (*Aloe vera* (L.) *Burm.*) [68] often act to promote wound healing through their various bioactive ingredients. However, researchers speculate that the content of plant peptides is low, and it is difficult to achieve effective treatment with peptides from plants alone. Therefore, new studies are seeking other species to broaden the range of skin treatments.

Animal peptides have attracted extensive attention from scholars as new molecular platforms for skin therapeutics [69]. For example, gecko duct analogs (GJ-CATH3) have been found to exhibit significant wound-healing properties in mouse models with full-thickness skin wounds. These peptides have the potential to stimulate HaCaT cell proliferation while also preventing a decrease in SOD activity and an increase in MDA concentration in damaged skin tissues [70]. Multifunctional peptides were found in salamander (*Cynops orientalis*) skin, and might play an important role in the host’s immune response to bacterial infection and skin wound repair [71,72].

Previous studies have shown that many aquatic animal peptides also have skin-healing properties. The relationship between wound healing and wound microbiome colonization was investigated by using skin collagen peptides of salmon (*Oncorhynchus keta*) and tilapia (*Oreochromis mossambicus*). Several studies revealed that collagen peptides were related to the regulation of microbial community colonization in wound tissue. They were also found to promote wound healing by controlling inflammatory response and increasing wound angiogenesis and collagen deposition [73,74]. Peptides extracted from the enzymatic digestion of perch (*Lateolabrax japonicus*) could also accelerate wound healing by enhancing the formation of microvessels at the wound site [75]. The active peptides (Aps) of pearl oyster (*Pinctada Martensii*) increased collagen synthesis and type III collagen content in wounds via the TGF-/Smad signaling pathway, inhibiting scar formation and promoting skin wound healing in [76].

It has been reported that amphibian skin has an excellent ability to enhance wound healing [77]. These active ingredients, especially polypeptide extracts, have been proven to effectively promote skin wound healing. Thus, polypeptide extracted from amphibian skin has great potential for skin repair [78]. According to Table 1, reporting on skin treatments with amphibian-sourced peptides, a new 24-residue peptide belonging to the ducting family was identified from the skin of the plateau frog (*Nanorana pleskei*) and has been shown to promote wound contraction and repair in in vivo and in vitro experiments [79]. Ot-WHP, as a wound-healing-promoting peptide from the Chinese concave-eared torrent frog (*Odorrana tormota*), has the same effect on skin treatment, according to another study [80]. There was also a new short peptide (named RL-QN15) in the skin secretions of *Rana serrata* that could regulate cytokines secreted by macrophages and accelerate re-epithelization and granulation tissue formation [81]. Researchers evaluated two peptides, called CW49 and pseudin-2 (Pse-T2), from frog (*Odorrana graham*) skin, showing that the former could treat skin injury by promoting angiogenesis [82], and the latter by destroying the membrane integrity to kill bacterial cells [83]. Peptides extracted from other frogs also had skin-healing properties [84,85,86,87,88,89,90,91,92,93].

### 5.2. Infectious and Inflammatory Skin Disease

Some peptides have been revealed to induce cell proliferation, migration, and differentiation. These peptides could also regulate inflammatory response and control the production of various cytokines/chemokines. These capabilities enable them to promote wound healing and improve skin barrier function [95]. An increasing body of evidence proves that peptides play an important role in skin defense, and some have therapeutic effects on viral resistance in atopic eczema (AE) [96]. A homing peptide CRKDKC (CRK) was found to be widely used in wound recovery and angiogenesis in tumors, and the removal of cysteine from CRK produces a skin-homing therapeutic molecule (DCN-tCRK). Experiments in vivo have proved that this molecule could inhibit TGF in skin-β signal transduction, thereby improving recessive dystrophic epidermolysis bullosa [97]. There is also an emollient containing oat plant extract that has revealed anti-inflammatory and barrier repair properties to treat specific dermatitis [98]. Other antimicrobial peptides had similar dermatitis repair effects [99].

### 5.3. Neoplastic Skin Disease

Emerging evidence suggested that peptide components not only treated some inflammatory skin diseases but played a role in the healing of skin tumors [100]. An example of this was three antimicrobial peptides which were identified from frog skin secretions by ‘shotgun’ cloning and MS/MS fragmentation. By testing the antimicrobial and biofilm activity of microorganisms, they were found to induce bacterial death by destroying cell membranes and binding to bacterial DNA, thereby alleviating skin cancer [101]. Antimicrobial peptides obtained from insects were shown to have antibacterial and anti-inflammatory effects on the skin, but also anti-cancer effects [102]. Another study found that the insect-derived peptide poecilocorisin-1 had a potential therapeutic effect on malignant melanoma skin cancer [103]. The combined therapy of antimicrobial peptides and chemotherapeutic drugs was also developed as a new method for the treatment of skin cancer, and revealed a synergistic therapeutic effect on skin cancer in mice [104].

## 6. Clinical Application and Prospect of Peptides in Skin Healing

Skin wound healing, especially chronic wound healing, is a common and challenging clinical problem. The development and utilization of clinical therapeutic agents have attracted extensive attention [105,106,107]. There is thus an urgent need to develop new interventions to promote skin repair. Recent studies showed that both peptides and nanoparticles might be potential therapies for skin wounds [108,109]. Researchers synthesized antibacterial photodynamic gold nanoparticles (AP-AuNPs), which combined an antibacterial peptide and nanoparticles. AP-AuNPs are used as a wound-dressing nanomaterial in skin infections to promote wound healing [110]. In addition, ZnO nanoparticles of different sizes were also functionalized with an amphipathic peptide to improve their photoprotection capability in skin [111]. For example, marine peptides extracted from tilapia (*Oreochromis mossambicus*) were combined with a biological material called chitosan (CS) and used as a therapeutic agent for skin wound healing [112]. In many studies, researchers designed new peptide scaffolds to obtain peptides that could be stable in the wound environment, and their efficacy in promoting wound healing was demonstrated in vitro and in vivo [113]. The fusion of peptides and mediators could also form potential new drugs for the treatment of skin wounds and inflammation. One study was assessed to analyze the molecular design characteristics of peptide-based hydrogels for improving wound healing [114]. Transglutaminase (TG) was thus identified from the transcriptomes of *Spirulina*, and its free-radical-scavenging potential was evaluated. With the help of an electrostatically spun chitosan/polyvinyl alcohol nanofiber pad, this TG could promote wound healing in vitro [115]. Therefore, we expect to see the design of a carrier that could carry peptides, maximize the efficacy of peptides, expand the application range of peptides, and develop more drugs for skin treatment [1]. 

## 7. Discussion

There have been many attempts to explore and develop peptide extraction and synthesis methods to advance beyond traditional models and improve peptide availability. Therefore, this review summarized these extraction, modification, and synthesis methods to provide innovative ideas for the efficient utilization of peptides. In addition, it focused on the peptides that have shown protective and therapeutic effects on the skin. Although the potential of peptides for skin therapy was found in many organisms, many peptides with skin therapeutic properties are yet to be discovered. The current literature collection suggests that peptides are relatively scarce in clinical applications. One possible reason is that peptides are not easily stored, due to their facility to break down and deteriorate easily. Therefore, it is necessary to explore new ways to facilitate their clinical application, especially regarding their advantages in the skin field which could provide relief for patients suffering from skin injuries. Based on experimental evidence, it was found that peptides could affect skin mechanisms by regulating inflammation, oxidative stress, apoptosis, aging, and autophagy. These findings suggest that peptides can influence many pathophysiological processes and biochemical signaling pathways. Therefore, peptides have the potential for use as therapeutic agents for other diseases. The findings provide a theoretical reference for broader applications of peptides.

## 8. Conclusions

This review describes the extraction, modification and synthesis of peptides in recent years, which might provide ideas for the acquisition and development of novel peptides. In addition, this review highlights possible mechanisms of action of peptides in the treatment of skin diseases, and these findings suggest peptides could be candidates for the alleviation and treatment of skin diseases. However, many detailed studies are needed to clarify whether peptides have the same effect in other pathologies.

## Figures and Tables

**Figure 1 molecules-28-00908-f001:**
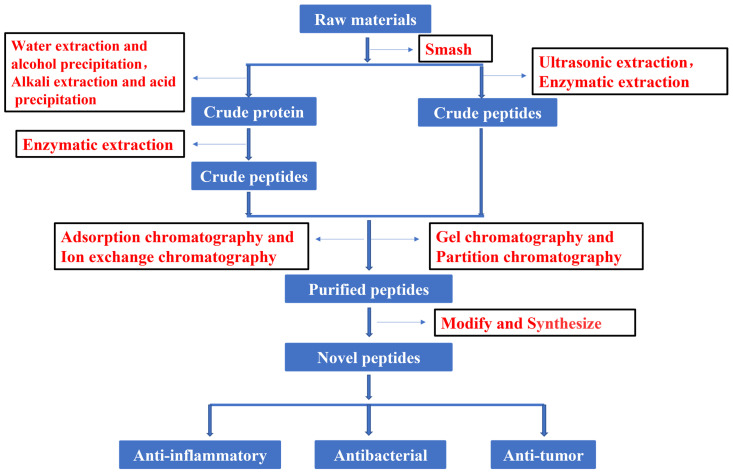
Flow chart of peptide process.

**Figure 2 molecules-28-00908-f002:**
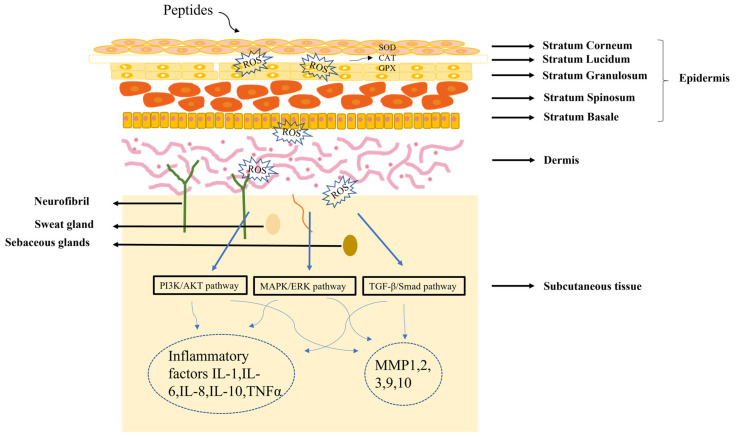
Diagram of the mechanism of peptide treatment of skin damage.

**Table 1 molecules-28-00908-t001:** Amphibians for skin treatment.

Species	Latin Name	Petite Nomenclature	Sequence	In Vitro	In Vivo	Disease	Quote
Plateau frog	*Nanorana pleskei*	Cathelicidin-NV	ARGKKECKDDRCRLLMKRGSFSYV	HACAT	Full-thickness wound in mice	Skin wound healing	[79]
The Chinese concave-eared frog	*Odorrana tormota*	Ot-WHP	ATAWDLGPHGIRPLRPIRIRPLCG	HACAT, RAW	Full-thickness wound in mice	Skin wound healing	[80]
Frog	*Rana Serrata*	*RL-QN15*	QNSYADLWCQFHYMC	HACAT, RAW	Full-thickness wounds in mice and oral ulcers in rats	Wound healing mouth ulcers	[81]
Frog	*Odorrana graham*	CW49	APFRMGICTTN	None	Full-thickness wound in diabetic mice	Diabetes skin wound healing	[82]
Frog	*Pseudis paradoxa*	Pse-T2	None	RBC, HACAT	Full-thickness wound in mice	Skin wound healing	[83]
Frog	*Odorrana grahami*	AH90	ATAWDFGPHGLLPIRPIRIRPLCG	HACAT, RAW	Full-thickness wound in mice	Skin wound healing	[84]
Frog	*Rana pleurade*	RP-1	None	HACAT	Irradiation-induced trauma in mice	Radiation-induced wound healing	[85]
Frog	*Rana pleurade*	RL-RL10	RLFKCWKKDS	HACAT	Full-thickness wound in mice	Skin wound healing	[86]
Frog	*Rana limnocharis*	OA-FF10	FFTTSCRSGC	HDPs	Full-thickness wound in mice	Skin wound healing	[87]
Frog	*Rana limnocharis*	cathelicidin-OA1	IGRDPTWSHLAASCLKCIFDDLPKTHN	HACAT, HDPs	Full-thickness wound in mice	Skin wound healing	[88]
Frog	*Rana pleurade*	chensinin-1	None	None	Full-thickness wound in mice	Skin wound healing	[94]
Frog	*Rana limnocharis*	OA-GL21	GLLSGHYGRVVSTQSGHYGRG	HACAT, HDPs	Full-thickness wound in mice	Skin wound healing	[90]
Frog	*Rana limnocharis*	OM-GL15	GLLSGHYGRASPVAC	None	Photodamage in mice	Skin photodamage	[91]
Frog	*Rana sierrae*	brevinin-1Ma	FLPILAGLAANLVPKLICSITKKC	None	None	Skin immune protection	[92]
Frog	*Odorrana livida*	antioxidin-RL	AMRLTYNRPCIYAT	HACAT, HDPs	Photoaging in mice	Skin photoaging	[93]

## Data Availability

Publicly available datasets were analyzed in this study. This data can be found here: https://pubmed.ncbi.nlm.nih.gov/.

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
