# Peer review of "Review on Extraction, Modification, and Synthesis of Natural Peptides and Their Beneficial Effects on Skin"

_molecules, 2023, doi:10.3390/molecules28020908_

Round 1
Reviewer 1 Report
The text seems to be a list of different researchs without a proper correlation between them.
Some citations do not seem related to the sentence in the manuscript. This is the case of citations 1, 12, 14 among others.
Figure 2. is a strange representation of the skin layers. Cuticle is not the most superficial layer of the skin. Leather layer is not clear what is. The different mechanisms represented in the figure are not explained clearly.
4.2. Skin disease is generic.
Reviewer 2 Report
In this review, the extraction, modifications, synthesis of peptides, and exploring the research progress of peptides in the field of skin wounding have been summarized. The knowledge provides a good foundation for the application and development of peptides in the future. Overall, the manuscript's ideas appear justified. Nevertheless, the manuscript has been organized a bit carelessly, and the following are some unsatisfying parts that need to be revised.
1. The number section should be arranged from the Introduction part (number 1) to the Discussion part (number 7). Where is the conclusion section?
2. All the references should be in only format. Therefore, it is better for the author to check Instructions for Authors ( https://www.mdpi.com/journal/molecules/instructions).
3. Line 65, What are the five common proteases for protein extraction? Please provide the exact source of protease.
4. Line 72, Figure 1. Flow chart of peptide process, the type and size of fonts should be unique.
5. Line 262-263: “Recent studies showed that both peptides and nanoparticles might be the potential therapies for skin wounding”. Was there any research combined between peptide and nanoparticles on the treatment of skin wounding?
Round 2
Reviewer 1 Report
You have updated the manuscript intensively according to my comments and the ones from another reviewer. Now it looks better.
You can modify the following:
In line, 183 authors say "We have found that peptides repair skin damage through a variety of mechanisms" including 2 references. Then this is not a contribution of the authors because it is a review, then better say: "different authors found ......
Author Response
Dear Reviewer:
We are glad that our responses were to your satisfaction. We appreciate you taking the time to review this manuscript and giving us your suggestions. We have corrected the manuscript according to your suggestion. Revised portion are marked in red in the paper. The main corrections in the paper and the responds to the reviewer’s comments are as flowing:
Point 1.Comment and Suggestion:
In line, 183 authors say "We have found that peptides repair skin damage through a variety of mechanisms" including 2 references. Then this is not a contribution of the authors because it is a review, then better say: "different authors found ......
Response 1:
Thank you very much for your thoughtful advice. We have amended this section as you requested (Line 183-185).
Best wishes for you.
Reviewer 2 Report
I agreed the revised mamnuscript
Author Response
Dear Reviewer:
We appreciate your kind suggestions and thank you very much for your approval of our responses.
Best wishes for you.